## Replications

psychology

face learning, face familiarity, EEG, ERP, N250

**Authors for correspondence:**
Werner Sommer
e-mail: werner.sommer@cms.hu-berlin.de
Katarzyna Stapor
e-mail: katarzyna.stapor@polsl.pl

# The N250 event-related potential as an index of face familiarity: a replication study

Werner Sommer[1], Katarzyna Stapor[2],
Grzegorz Kończak[4], Krzysztof Kotowski[2], Piotr Fabian[3],
Jeremi Ochab[5], Anna Bereś[6] and Grażyna Ślusarczyk[7]

[1]Department of Psychology, Humboldt-University at Berlin, Rudower Chaussee 18, 12489 Berlin, Germany
[2]Department of Applied Informatics, and [3]Department of Algorithmics and Software, Silesian University of Technology, Gliwice, Poland
[4]Department of Statistics, Econometrics and Mathematics, University of Economics in Katowice, Poland
[5]Department of Theory of Complex Systems, [6]Department of Cognitive Neuroscience and Neuroergonomics, and [7]Department of Design and Computer Graphics, Jagiellonian University, Krakow, Poland

WS, 0000-0001-5266-3445; KS, 0000-0003-3003-6592; KK, 0000-0003-2596-6517; JO, 0000-0002-7281-1852

The neural correlates of face individuation—the acquisition of memory representations for novel faces—have been studied only in coarse detail and disregarding individual differences between learners. In their seminal study, Tanaka *et al.* (Tanaka *et al.* 2006 *J. Cogn. Neurosci.* **18**, 1488–1497. (doi:10.1162/jocn.2006.18.9.1488)) required the identification of a particular novel face across 70 trials and found that the N250 component in the EEG event-related potentials became more negative from the first to the second half of the experiment, where it reached a similar amplitude as a well-known face. We were unable to directly replicate this finding in our study when we used the original split of trials. However, when we applied a different split of trials we observed very similar changes in N250 amplitude. We conclude that the N250 component is indeed sensitive to the build-up of a robust representation of a face in memory; the time course of this process appears to vary as a function of variables that may be determined in future research.

## 1. Introduction

The recognition of the faces of individual persons is crucial for social life. Although it is often very easy to recognize familiar faces, it is much harder for unfamiliar faces [1] especially from single instances [2]. Therefore, the development of stable traces

of faces over repeated encounters despite image variability is being seen as an outstanding research question [3].

Event-related potentials (ERPs), derived from the EEG, provide a number of components that are sensitive to the cognitive processing of faces. Following the P1 component, which is mainly sensitive to domain-general visual processes, faces elicit a prominent N1 component peaking around 170 ms post-stimulus onset (N170; [4]). The N170 is larger to faces than to most other objects and increases in amplitude and latency when the structure of a face is hard to perceive; however, it has rarely been found to be sensitive to individual faces or to face familiarity [5–7].

In contrast with the N170, the subsequent N250 component has been shown to be familiarity-sensitive. The N250 is negative at occipito-temporal sites and positive at frontal sites and is thought to be generated in or near the fusiform gyrus [7,8]. The N250 increases if a face image is the same as an immediately preceding face as compared with when it is different [9]. Importantly, this early repetition or priming effect—also termed N250r—is more pronounced for familiar faces than for unfamiliar faces. With respect to face individuation, the N250r has been shown to increase when a face grows more familiar [5]. Other studies showed that also the N250 component itself increases when unfamiliar faces are presented repeatedly [6] and that similar effects hold when experts view non-face objects in their domain of expertise [10]. Supporting the findings from [6], authors of [11] showed larger N250 amplitudes to explicitly learned than to novel faces. Authors of [3] reported that the increase of N250 holds also when during the ERP test different images were used than during (incidental) learning. Hence, it has been concluded that the N250 is associated with the processing of familiar objects at the subordinate representational level [6,12,13].

Previous studies, demonstrating the relationship of the N250 with face (or object) familiarity, have either compared familiar with (different) unfamiliar faces or familiar and unfamiliar objects (e.g. cars, dogs; [10]) or they have compared the N250 across two or three consecutive blocks of trials [5,6,14] as stimuli increased in familiarity. Authors of [7] presented (several) faces in four blocks which had been seen in videos before and found an increase of N250 after the first block. Furthermore, previous studies disregard individual differences which are prominent in face cognition (e.g. [15]).

The starting point of the present study was the Joe/No Joe task of [6], where 11 unfamiliar faces and the participant's own face were presented multiple times. One of the unfamiliar faces was designated as 'Joe's' face (target face). When the responses to 'Joe' presentations were averaged separately for the first half and the second half of the experiment (35 trials each), the N250 amplitude within a region of interest (ROI) of 12 electrodes at posterior areas of the scalp in the time range 230–320 ms post-stimulus was larger (more negative) in the second than in the first half of the experiment and became indistinguishable from the N250 to the highly familiar picture of the participant's own face. In the present study, we attempted to replicate the results of the seminal study of Tanaka *et al.* [6].

# 2. Methods

This article received results-blind in-principle acceptance (IPA) at Royal Society Open Science. Following IPA, the accepted Stage 1 version of the manuscript, not including results and discussion, was preregistered on the OSF (https://osf.io/j947u/). This preregistration was performed after data analysis. The raw EEG recordings in Brain Vision format, scripts for preprocessing of the raw recordings, and the code for generating results are available at https://osf.io/7x6w5.

## 2.1. Participants

Twenty participants were recruited among Polish students of Jagiellonian University, Krakow. All participants were healthy, right-handed Caucasians and had normal or corrected-to-normal vision. They signed informed consent and received a reward for participation. The study was approved by the Ethics Committee of the Institute of Applied Psychology at the Jagiellonian University, Krakow, Poland (25 June 2018).

Data of four participants were rejected; two, because they finished only part of the experiment, one because of very low response accuracy (<60%), and one because of the low signal-to-noise ratio of the ERP signals. The final sample consisted of 16 participants (12 females; mean age = 21.5 years; range: 19–23).

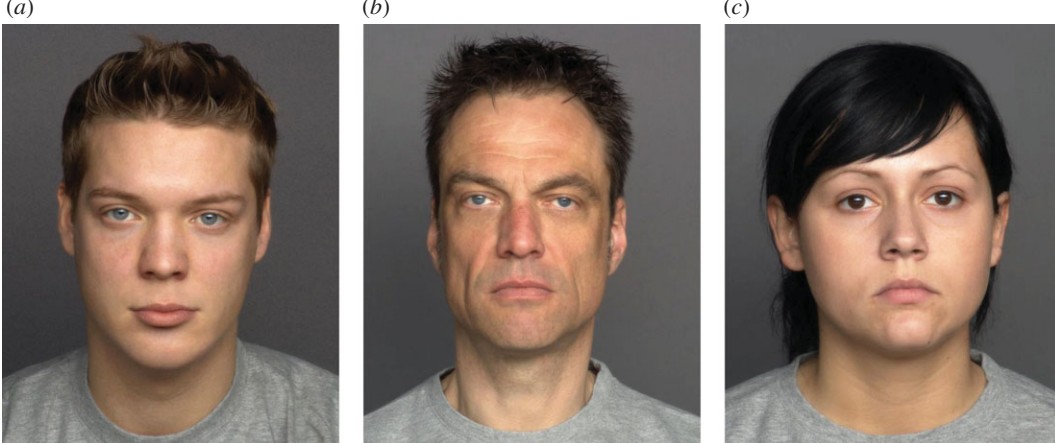

**Figure 1.** Sample faces from the FACES Lifespan Database for Facial Expressions [16] as used in the present study.

## 2.2. Stimuli

Two sets of 11 colour portraits each of male or female young adult European individuals were used as stimuli. Pictures were taken from the FACES Lifespan Database for Facial Expressions [16]. Sample faces from this database are presented in figure 1. One of the faces in each set was randomly selected as a target ('Joe' or 'Jane') for all participants of the corresponding gender. A picture of the participant (the Own face) was taken right before the experiment on a background similar to the one in FACES pictures. It was manually cropped and normalized to the same dimensions as the other pictures (335 × 419 pixels), using GIMP software. The subset of pictures used for a given participant was gender matched.

## 2.3. Procedure

After signing written informed consent and reading the task instructions, participants were seated in a dimly lit, quiet and electrically shielded room. A computer screen was placed at a viewing distance of 80 cm. Participants were instructed to sit still, keep quiet and maintain central eye fixation during the trials.

The experimental procedure was a replication of the Joe/no Joe (or Jane/no Jane) task of Tanaka *et al.* [6]. Participants were instructed to monitor the centrally presented faces, where the centre of the nose was always in the centre of the screen, and indicate whether the face was the target or not by pressing a right or left button using the index and middle finger of the right hand.

At first, participants were familiarized with the Joe or Jane target face. Although this part is crucial for the process of learning, the original description by Tanaka *et al.* [6] is a bit vague: 'Subjects were then shown the target Joe (Jane) face, and asked to study it.' (p. 1490). Thus, we decided to present the Joe (Jane) face on the screen right before the experiment, for 10–60 s depending on the participant's needs. Although it limits the possibility to analyse the initial phase of face learning, we decided to stick with the (presumably) original procedure. One difference from the original procedure is that we used the same Joe/Jane face for all male/female participants to avoid additional variability connected with physical face characteristics.

The experiment was implemented with E-Prime software. Faces covered around 3.0° × 4.8° visual angle. Each trial began with a 500 ms presentation of a white fixation cross. Then, a face from the set matching the participant's gender was presented for 500 ms, after which the screen went blank for 500 ms; finally, the question 'Joe?' ('Jane?' for female participants) was displayed for 500 ms. Participants were instructed to respond only after the question was displayed on the screen. Trials with responses preceding the question were excluded from analysis; if more than one response was given, only the first one was considered. No feedback about the performance was given.

Each of the 36 experimental blocks (one more than in [6]) consisted of 24 trials (two presentations each of 10 Other faces, two presentations of the Joe/Jane face and two presentations of the participant's Own face). The order of stimuli was pseudo-randomized and different in each block and participant. Joe/Jane and Own faces were never repeated immediately to avoid repetition effects [14]. This presentation regime implies that the intervals (intervening Own and Other faces) between Joe/Jane faces during the

experiment varied across participants. In total, there were 864 trials per participant (72 Joe trials, 72 Own trials, plus 72 times 10 Other trials) with self-paced breaks between blocks. In total, the task took about 30 min.

## 2.4. EEG/ERP methods

The EEG signal was recorded using the BrainProducts ActiChamp amplifier with 2500 Hz sampling frequency. There were no additional filters applied during recording (the device itself has bandwidth DC–7500 Hz). The BrainProducts ActiCap was used with 64 active electrodes located according to the 10–10 standard [17] but oversampling the posterior scalp regions. The Cz electrode was used as an initial common reference. The electrooculogram (EOG) was recorded from a passive electrode located at the outer canthus of the left eye. All impedances were kept below 10 kΩ. As suggested by the amplifier manufacturer, active noise cancelling was deactivated because active electrodes were used in an electrically shielded room. A Chronos multifunctional response and stimulus device (Psychology Software Tools, USA) was used to collect responses from participants and correct stimulus display times according to the markers from a photodiode mounted on the screen (differences up to 15 ms).

Offline, corrupted channels (three channels at most) were replaced by spherical spline interpolation [18]. The EEG signals were digitally low-pass filtered at 40 Hz, using third-order zero-phase forward-backward digital Butterworth filtering [19] with a slope of 6 dB, re-calculated to a common average reference and down-sampled to 250 Hz. As the electrodes in our set-up were not evenly distributed on the scalp (20 in the anterior and 44 in the posterior scalp), the common average for all electrodes was calculated on a subset of 31 evenly distributed electrodes (Fp1, Fp2, F7, F3, Fz, F4, F8, FT9, FC5, FC1, FC2, FC6, FT10, T7, C3, C4, T8, CP5, CP1, CP2, CP6, TP9, P7, P3, Pz, P4, P8, TP10, O1, Oz, O2).

The EOG was filtered using independent component analysis by finding and removing components that were highly correlated with the EOG signal [20]. Epochs were extracted for time windows between 100 ms before and 900 ms after stimulus onset and corrected with respect to 100 ms pre-stimulus baselines. Trials with activity ranges greater than 100 μV within any channel or with incorrect responses were discarded from further analyses. All operations were performed using Python 3 and the MNE package [21].

# 3. Results

## 3.1. Behavioural results

The average accuracy of responses for the 18 participants that completed the whole experiment (including two participants rejected for ERP analyses alone) was high ($M = 91.45\%$, s.d. = 11.47%) with significantly higher values in the second half of the experiment ($M = 95.66\%$, s.d. = 5.76%) than in the first half ($M = 87.25\%$, s.d. = 18.99%) according to the Wilcoxon signed-ranked test ($Z = 3.03$, $p = 0.002$). The non-parametric test was used because the normal distribution, required for repeated-measures analysis of variance (ANOVA) was violated according to the Shapiro–Wilk test ($p = 0.034$). As compared with the first half of the experiment, in the second half, most participants (except for three) achieved greater or equal accuracy for the Joe/Jane faces (table 1). Average accuracies for both Joe/Jane (95.11%) as well as Own (93.98%) faces were significantly higher than for Other faces (90.84%; versus Joe/Jane: $Z = 2.72$, $p = 0.007$; versus Own: $Z = 3.29$, $p < 0.001$). Mean reaction times (for correct trials only) over all conditions were significantly shorter in the second than first half ($M = 1142$ versus 1186 ms, s.d. = 102 versus 78 ms) of the experiment ($Z = 3.07$, $p = 0.002$). These behavioural results confirm the findings of Tanaka *et al.* [6] and show that participants effectively learned during the experiment, as a group but also in most individual cases.

## 3.2. ERP results

### 3.2.1. Direct replication

Following [6], two ROIs with six electrodes each were selected for measuring N250 amplitude in the right and left hemispheres (TP10, P8, P10, PO8, PO10, O2 and TP9, P7, P9, PO7, PO9, O1). ERP waveforms were obtained by averaging over all electrodes within each ROI, per condition (Joe/Jane, Own, Other) and each half of the experiment (figure 2). The resulting ERP waveforms resembled the ones shown

**Table 1.** Accuracy of responses to Joe/Jane faces (in %) for 18 of 20 participants.

|  | 1 | 2 | 3 | 4 | 5 | 6 | 7 | 9 | 11 | 12 | 13 | 14 | 15 | 16 | 17 | 18 | 19 | 20 |
|---|---|---|---|---|---|---|---|---|---|---|---|---|---|---|---|---|---|---|
| 1st | 95 | 100 | 100 | 100 | 100 | 94 | 94 | 100 | 100 | 92 | 97 | 81 | 97 | 36 | 97 | 97 | 89 | 90 |
| 2nd | 97 | 97 | 94 | 100 | 100 | 100 | 97 | 100 | 100 | 100 | 100 | 100 | 100 | 94 | 100 | 100 | 100 | 85 |
| avg | 96 | 99 | 97 | 100 | 100 | 97 | 95 | 100 | 100 | 96 | 99 | 90 | 99 | 65 | 99 | 99 | 94 | 88 |

1st and 2nd: first and second half of experiment; avg.: all trials.

in (fig. 3 in [6]), especially for Other faces, but they also show some differences. For present purposes, the most important difference is that the N250 'dip' was visible only for ERPs to Joe/Jane faces but not at all for ERPs to Own faces. Furthermore, this dip was present in both halves of the experiment and not just in the second half as in [6]. In addition, there was a negative-going shift in the ERPs between 400 and 600 ms post-stimulus that was not present in the waveforms shown in [6].

As in [6], the average amplitude of the N250 was measured in the averaged ERPs in each ROI between 230 and 320 ms and submitted to a repeated-measures ANOVA with factors Condition (Own, Joe, Other faces), Experiment Half (First, Second) and Hemisphere. Degrees of freedom were adjusted for sphericity violations according to the conservative Greenhouse–Geisser procedure [22].

Replicating Tanaka *et al.*'s results, there were clear N250 amplitude differences between conditions, $F(2,15) = 14.06$, MSE $= 7.84$, $p < 0.001$. However, confirming visual impressions, condition effects were not qualified by an interaction with experiment half ($F(2,15) = 0.91$, MSE $= 1.89$, $p = 0.38$). More specifically, the null hypothesis of equal N250 amplitudes cannot be rejected at the $\alpha = 0.01$ level according to Bonferroni-corrected follow-up contrasts. The strip plots and distributions of the differences in figure 4 show the detailed interaction of condition and experiment part. The contrasts revealed also that N250 to Joe faces significantly differed from Own faces in both parts of the experiment, whereas in [6] amplitudes had been indistinguishable in the second half. Furthermore, we could not replicate the left-hemispheric asymmetry reported in [6], $F(1,15) = 2.23$, MSE $= 8.67$, $p = 0.16$.

### 3.2.2. Exploring an alternative trial division—1/3 to 2/3

We searched for an alternative split by checking the difference in means between experimental parts in steps of 1/24 of trials (i.e. 1/24 versus 23/24, 2/24 versus 22/24, etc.). Only the divisions providing a reasonable number of trials (at least 20 per participant [23]) for grand ERP averages to be meaningful were considered. In the second step, we chose from the candidates the split giving the maximum difference of means. It turned out that split 1/3–2/3 was the best over all the splits explored. Grand averages of this new division are shown in figure 3.

ANOVA of N250 amplitude with the new division (factor Experiment part (first 1/3 versus last 2/3 of trials) revealed a significant interaction of condition and experiment part, $F(2,15) = 5.73$, MSE $= 1.56$, $p = 0.014$. Follow-up contrasts (Bonferroni corrected for multiple comparisons, $p < 0.01$) revealed that N250 amplitudes for targets (Joe or Jane faces) were significantly more negative in the last 2/3 than in the first 1/3 of trials ($M = -0.51$ versus $0.28\,\mu V$) of the experiment (figure 3). The strip plots and distributions of the differences in figure 4 show the detailed interaction of condition and part.

Using the alternative division allowed to replicate the crucial effect of more negative N250 amplitudes to Joe/Jane faces in the later part of the learning process. These results indicate that similar learning effects were present as in [6] but shifted towards earlier parts of the experiment.

It is worth mentioning that the same analysis for N170 component in the time range of 130–200 ms did not reveal any significant interaction of condition and experiment part, supporting the notion of N250 as a specific familiarity effect and confirming the findings from the referenced articles.

## 4. Discussion

The present study aimed to replicate the findings from [6] that the N250 increases in amplitude from the first to the second part of a long series of face presentations, leading to increasing familiarity. In our replication, we monitored ERPs while participants viewed faces including their own face (familiar face), an experimenter-specified unfamiliar target face (Joe or Jane), and 11 other unfamiliar non-target faces. The increasing N250 amplitudes in averaged ERPs between two parts of the experiment, reported in [6] was observed only after dividing the target trials into the first 1/3 and the last 2/3 of samples, rather

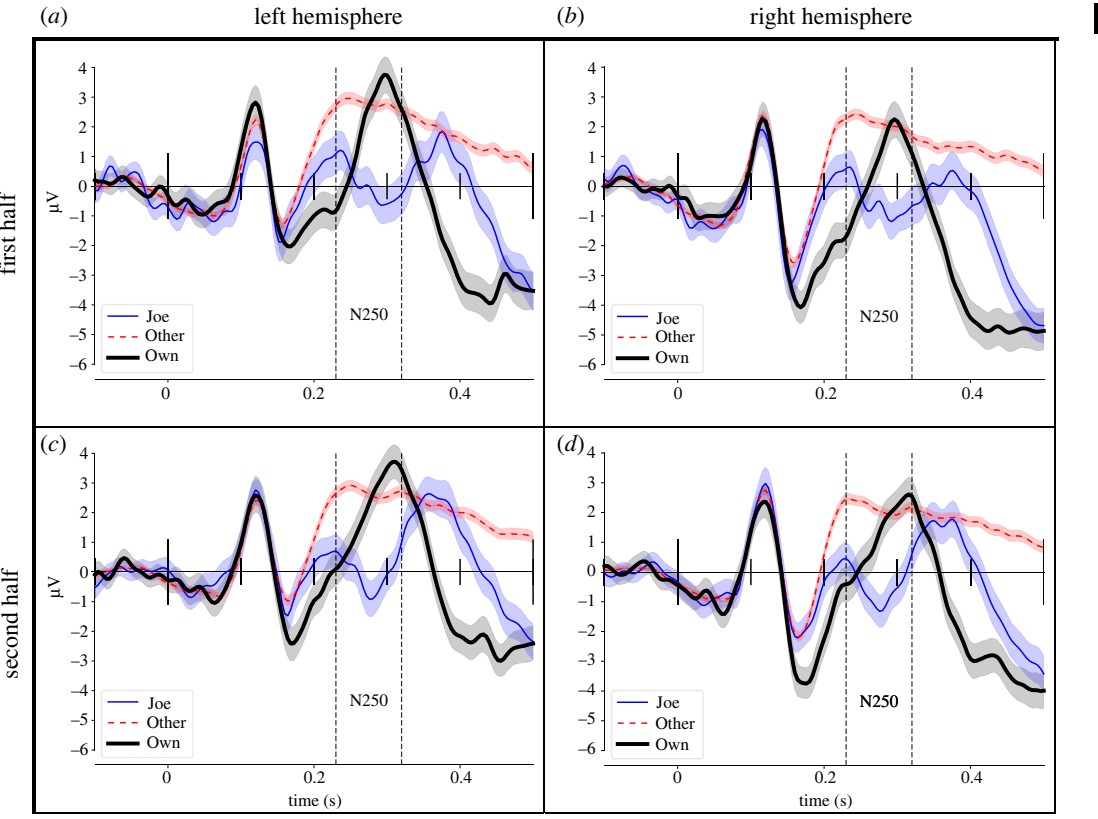

**Figure 2.** Grand-average ERPs for the two halves of the experiment averaged across channels at left and right regions of interest (the equivalent of fig. 1 from [6]).

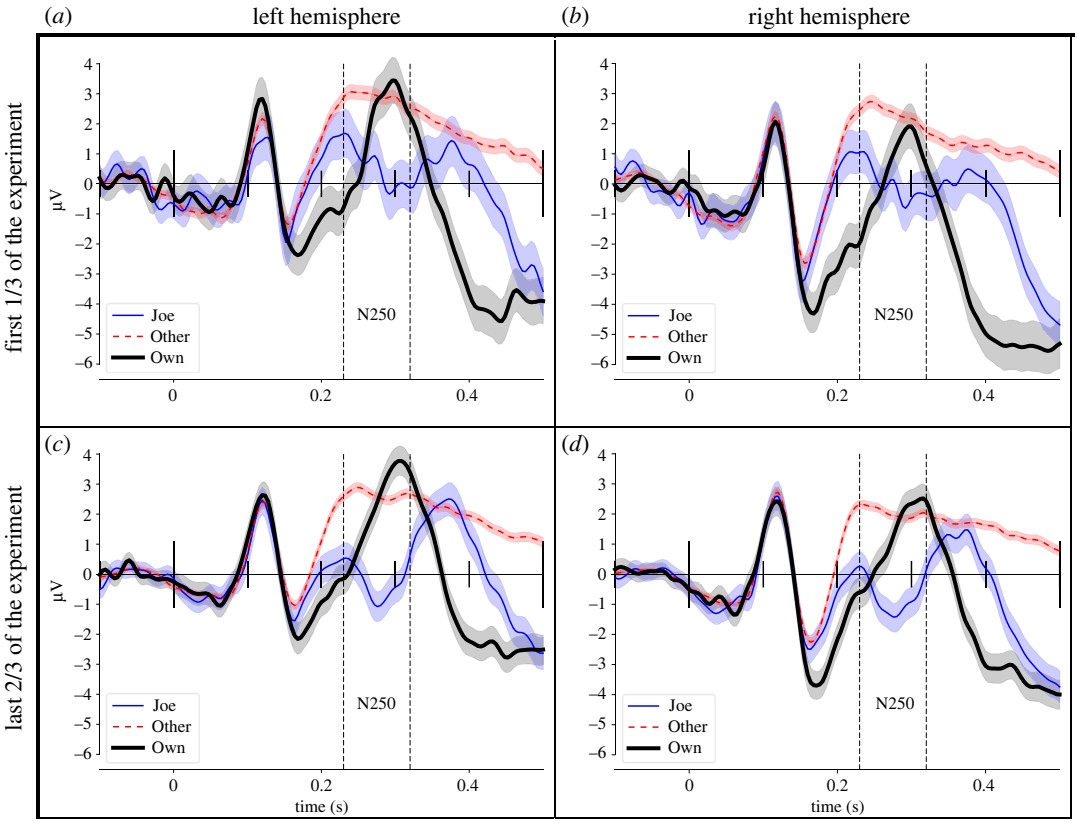

**Figure 3.** Grand-average ERPs for the first 1/3 of trials (*a,b*) and last 2/3 of trials (*c,d*) of the experiment for the left and right regions of interest of the N250 component (marked with vertical lines).

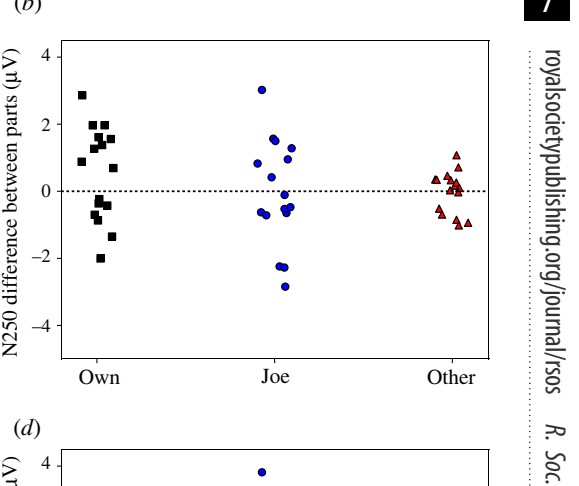

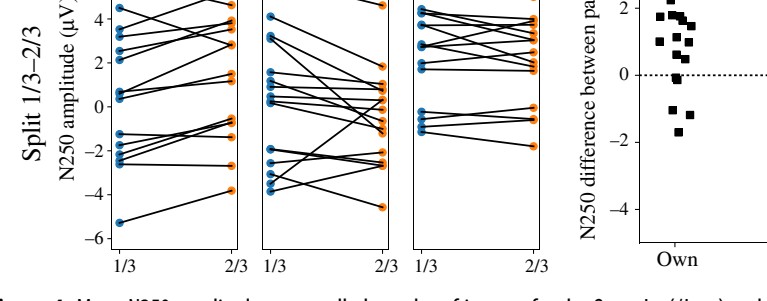

**Figure 4.** Mean N250 amplitudes across all electrodes of interest for the Own, Joe(/Jane) and Other conditions and their changes for different splits of the experimental trials; (a,c) show strip charts of linked observations for each participant for different splits; (b,d) show a distribution of N250 amplitude differences between parts. The asterisk indicates a reliable difference between the two parts of the experiment for the Joe condition.

than the first and second half. This may suggest that, on average, our participants learned faster, our Joe and Jane faces were easier to learn, and/or the pre-learning phase was more efficient than in [6]. We contacted J. Tanaka about the pre-learning regime in [6], and it seems that, similar to the present regime, they showed the Joe/Jane face to the participant as long as they wanted. Therefore, controlling the pre-experimental exposure of the target face and other variables known to affect face memory appears to be desirable in future studies on the acquisition of representations of faces.

Whatever the reasons, why the overall difference between early and late trials emerged earlier in the present study as compared with the report by [6], it does lend support to the idea that N250 amplitude is sensitive to the acquisition of memory representations of a new face image. The N250 amplitude for Other faces was stable across both parts of the experiment. The results for the non-Joe faces indicate that the ERP changes to target faces were no artefact of time-on-task such as fatigue but instead were indeed due to learning.

A more realistic face learning procedure would involve the recognition of the 'Joe/Jane' face across a larger range of image variations of the same person (e.g. [7]); this may be especially relevant as it is surprisingly difficult to recognize unfamiliar faces if different views of the same faces are used—as unavoidable in real-life settings [24]. The findings from unfamiliar-face-matching described in [1] show that people have a perceptual problem with recognizing that different images can represent the same unfamiliar face identity. Therefore, it would be a promising extension of the present approach to apply the present paradigm to unfamiliar face learning across image variability in future research. Another interesting line of research in the future might be a more sophisticated analysis of the participant's individual time courses of the development of N250 amplitude, especially in their relationship to face recognition abilities.

Apart from apparently faster learning of the target faces in the present study as compared with [6], we also found that Own faces did not elicit comparable N250 amplitudes as the target faces in the second part of the experiment. It is not clear how to explain this discrepancy. The Own faces in the present study had been taken immediately prior to the experiment and may have been relatively unfamiliar for the participants; however, it is not described how Tanaka et al. obtained the pictures for their own condition.

In conclusion, the present study replicated the amplitude increase of the N250 with increasing presentation repetitions of a target face. That the time course appeared to be faster in the present experiment than in the original study [6] does not call into question that the increase of N250 amplitude across the experiment supports the idea that this component represents the build-up of a stable memory trace for the target face.

Data accessibility. The raw EEG recordings in Brain Vision format, scripts for preprocessing of the raw recordings, and the code for generating results are available at https://osf.io/7x6w5.

Authors' contributions. W.S. advised data analysis and edited paper. K.S. designed study, analysed data, wrote the paper and supervised the study. G.K. designed the study, analysed data and wrote paper. K.K. developed code, wrote the paper and conducted experiment. P.F. pilot study, initial code and technical support. J.O. technical support, conducted the experiment and recruited participants. A.B. technical support, psychological supervision, conducted experiment and recruited participants. G.Ś. recruited participants, edited paper and organized the experiment.

Competing interests. We declare no conflict of interest.

Funding. This research was supported by statutory funds of Department of Applied Informatics, Silesian University of Technology, Gliwice, Poland (grant no. SUBB/RAu7/2020). J.O. was supported by the National Science Centre (Poland), grant no. DEC-2015/17/D/ST2/03492.

Acknowledgements. We acknowledge support by the German Research Foundation (DFG) and the Open Access Publication Fund of Humboldt-Universität zu Berlin.

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
