## [Peer Review File · Royal Society Open Science]

Review History

RSOS-202356.R0 (Original submission)

Review form: Reviewer 1

Do you have any ethical concerns with this paper?

No

Have you any concerns about statistical analyses in this paper?

Yes

Recommendation?

Major revision

Comments to the Author(s)

In the proposed experimental plan, I did not see a data analysis section. It is not stated how the ERP components (e.g, P1, N170, N250) be identified and analyzed. Please indicate whether there are plans to make the data available.

Review form: Reviewer 2

Do you have any ethical concerns with this paper?

No

Have you any concerns about statistical analyses in this paper?

No

Recommendation?

Accept with minor revision

Comments to the Author(s)

The research that the authors propose to replicate is indeed a hallmark piece of work in the area of face processing, and it would indeed be useful to see these findings on learning-based effects on the N250 component successfully replicated. The Introduction could be a bit more specific about the reasons why replication of this finding is particularly important, and why they chose to focus on the Tanaka et al. (2006) study but I think this can be easily justified. The Methods and proposed analyses are described clearly. I only have one specific concern about the proposed design. In the Methods section, the authors refer to a design aspect that was not described clearly in the original study:

"At first, participants were familiarized with the Joe or Jane target face. Although this part is crucial for the process of learning, the original description by Tanaka et al. (2006) [6] is a bit vague: "Subjects were then shown the target Joe (Jane) face, and asked to study it." (P. 1490). Thus, we decided to present the Joe (Jane) face on the screen right before the experiment, for 10 to 60 seconds depending on the participant's needs. Although it limits the possibility to analyze the initial phase of face learning, we decided to stick with the (presumably) original procedure."

I agree with the authors that details of the initial learning phase are crucial for any replication of a learning-based effect, and wonder why they did not contact the lead author of the original study (Jim Tanaka) for clarification. I am sure that Jim would be happy to provide them with all details required to match their learning phase with that of the original study. If data collection has not yet commenced, I recommend that the authors contact Dr Tanaka. If data collection has already happened, they should still do so and refer to any differences in learning procedures in their article.

Review form: Reviewer 3

Do you have any ethical concerns with this paper?

No

Have you any concerns about statistical analyses in this paper?

No

Recommendation?

Accept in principle

Comments to the Author(s)

This replication study aimed to test the reproducibility of a significant work in face perception literature that the acquisition of newly learned face representations could be indexed by the N250

ERP component (Tanaka et al., 2006). Also, the study was to investigate individual differences during the learning process. The methodology of the proposal is logically and technically sound.

Decision letter (RSOS-202356.R0)

Dear Dr Sommer

On behalf of the Editors, I am pleased to inform you that your Manuscript RSOS-202356 entitled "The N250 Event-related Potential as an Index of Face Familiarity: A Replication Study" deemed suitable for in-principle acceptance in Royal Society Open Science subject to minor revision in accordance with the referee and editor suggestions. Please find their comments at the end of this email.

The reviewers and handling editors have recommended publication, but also suggest some minor revisions to your manuscript. Therefore, I invite you to respond to the comments and revise your manuscript.

Please you submit the revised version of your manuscript within 7 days (i.e. by the 17-Feb-2021). If you do not think you will be able to meet this date please let me know immediately.

When submitting your revised manuscript, you will be able to respond to the comments made by the referees and upload a file "Response to Referees" in the "File Upload" step. You can use this to document any changes you make to the original manuscript. In order to expedite the processing of the revised manuscript, please be as specific as possible in your response to the referees.

Full author guidelines can be found here <https://royalsocietypublishing.org/rsos/replication-studies#AuthorsGuidance>.

Kind regards,
Professor Chris Chambers
Royal Society Open Science
openscience@royalsociety.org

on behalf of Professor Chris Chambers (Registered Reports Editor, Royal Society Open Science)
openscience@royalsociety.org

Associate Editor Comments to Author (Professor Chris Chambers):

Associate Editor: 1

Comments to the Author:

Three expert reviewers have now assessed the Stage 1 manuscript. The evaluations are positive, with all reviewers judging that both Stage 1 primary criteria are met. The reviewers do, however, recommend revisions to increase detail of the analysis plans (Reviewer 1) and clarity of the methods to determine how closely the procedures of the replication study match those of the original (Reviewer 2). Provided the authors are able to respond comprehensively to these points, in-principle acceptance should be forthcoming without requiring further in-depth Stage 1 review.

Reviewers' comments to Author:

Reviewer: 1

Comments to the Author(s)

In the proposed experimental plan, I did not see a data analysis section. It is not stated how the ERP components (e.g, P1, N170, N250) be identified and analyzed. Please indicate whether there are plans to make the data available.

Reviewer: 2

Comments to the Author(s)

The research that the authors propose to replicate is indeed a hallmark piece of work in the area of face processing, and it would indeed be useful to see these findings on learning-based effects on the N250 component successfully replicated. The Introduction could be a bit more specific about the reasons why replication of this finding is particularly important, and why they chose to focus on the Tanaka et al. (2006) study but I think this can be easily justified. The Methods and proposed analyses are described clearly. I only have one specific concern about the proposed design. In the Methods section, the authors refer to a design aspect that was not described clearly in the original study:

"At first, participants were familiarized with the Joe or Jane target face. Although this part is crucial for the process of learning, the original description by Tanaka et al. (2006) [6] is a bit vague: "Subjects were then shown the target Joe (Jane) face, and asked to study it." (P. 1490). Thus, we decided to present the Joe (Jane) face on the screen right before the experiment, for 10 to 60 seconds depending on the participant's needs. Although it limits the possibility to analyze the initial phase of face learning, we decided to stick with the (presumably) original procedure."

I agree with the authors that details of the initial learning phase are crucial for any replication of a learning-based effect, and wonder why they did not contact the lead author of the original study (Jim Tanaka) for clarification. I am sure that Jim would be happy to provide them with all details required to match their learning phase with that of the original study. If data collection has not yet commenced, I recommend that the authors contact Dr Tanaka. If data collection has already happened, they should still do so and refer to any differences in learning procedures in their article.

Reviewer: 3

Comments to the Author(s)

This replication study aimed to test the reproducibility of a significant work in face perception literature that the acquisition of newly learned face representations could be indexed by the N250 ERP component (Tanaka et al., 2006). Also, the study was to investigate individual differences during the learning process. The methodology of the proposal is logically and technically sound.

Author's Response to Decision Letter for (RSOS-202356.R0)

See Appendix A.

Decision letter (RSOS-202356.R1)

Dear Dr Sommer

On behalf of the Editor, I am pleased to inform you that your Manuscript RSOS-202356.R1 entitled "The N250 Event-related Potential as an Index of Face Familiarity: A Replication Study" has been accepted in principle for publication in Royal Society Open Science.

You may now progress to Stage 2 and complete the study as approved.

Please note that you must now register your approved protocol on the Open Science Framework (<https://osf.io/rr>), using the 'Submit your approved Registered Report' option and then the 'Registered Report Protocol Preregistration' option. Please use the Registered Report option even though your article is being accepted as a Stage 1 Replication. Further into the registration process, in the Journal Title field enter 'Royal Society Open Science (Replication article type, Results-Blind track)'. Please note that a time-stamped, independent registration of the protocol is mandatory under journal policy, and manuscripts that do not conform to this requirement cannot be considered at Stage 2. The protocol should be registered unchanged from its current approved state. Please include a URL to the protocol in your Stage 2 manuscript, and because you submitted via the Results-Blind track please note in the manuscript that the pre-registration was performed after data analysis (e.g. 'This article received results-blind in-principle acceptance (IPA) at Royal Society Open Science. Following IPA, the accepted Stage 1 version of the manuscript, not including results and discussion, was preregistered on the OSF (URL). This preregistration was performed after data analysis.')

Following completion of the research, we invite you to resubmit your paper for peer review as a Stage 2 Replication. Please note that your manuscript can still be rejected for publication at Stage 2 if the Editors consider any of the following conditions to be met:

- The Introduction and methods deviated from the approved Stage 1 submission (required).
- The authors' conclusions were not considered justified given the data.

We encourage you to read the complete guidelines for authors concerning Stage 2 submissions at: <https://royalsocietypublishing.org/rsos/replication-studies#AuthorsGuidance>. Please especially note the requirements for data sharing and that withdrawing your manuscript will result in publication of a Withdrawn Registration.

Once again, thank you for submitting your manuscript to Royal Society Open Science and I look forward to receiving your Stage 2 submission. If you have any questions at all, please do not hesitate to get in touch. We look forward to hearing from you shortly with the anticipated submission date for your stage two manuscript.

on behalf of Professor Chris Chambers (Registered Reports Editor, Royal Society Open Science)
openscience@royalsociety.org

Author's Response to Decision Letter for (RSOS-202356.R1)

See Appendix B.

RSOS-202356.R2 (Revision)

Review form: Reviewer 2

Do you have any ethical concerns with this paper?

No

Have you any concerns about statistical analyses in this paper?

No

Recommendation?

Accept with minor revision

Comments to the Author(s)

This is an interesting and generally well-written replication attempt, which did reproduce the main findings of Tanaka et al. (2006), with the important caveat that the time course of face learning was different.

I have some minor comments for consideration by the authors.

- 1) The pattern observed for the N250 for Own faces appeared slightly different from the Tanaka study. This is noted by the authors, but no explanation is provided. Judging from their Figures, it looks as if this N250 is triggered earlier than the N250 to Joe, and this could be mentioned. Some evidence for such a latency difference was also present in Tanaka et al., see their Figure 2, in particular over the right hemisphere, although I'm not sure that this was actually discussed in their article.
- 2) Could samples of the faces images used be included in this report? Tanaka et al. provided such samples in their Figure 1, and it would be informative if these images could in principle be compared.
- 3) Could some explanation be provided as to whether the Tanaka et al. study was selected among the many other N250 studies of face recognition as the target of this replication study? Because it was the first/most widely cited?
- 4) Typo in the abstract (line 13): spilt -> split

Review form: Reviewer 3

Do you have any ethical concerns with this paper?

No

Have you any concerns about statistical analyses in this paper?

No

Recommendation?

Accept with minor revision

Comments to the Author(s)

This study tried to replicate the work of Tanaka et al. (2006) which found that the N250 component increased for an experimentally familiarized face in the second half of the experiment relative to the first half. Although this study could not directly replicate the previous findings by dividing the trials into two equal halves, it did replicate similar N250 increase when comparing 1/3 vs. 2/3 trials suggesting that participants may learn faster in this study. The methodology was sound, the results were clear, and the explanations were convinced. However, since another aim of this study was to investigate the individual differences during the learning process, I suggest the authors to give more exploration and discussion of this part of results.

Main points:

1. What conclusion could be drawn based on the present individual data of the N250 results?
2. Are the individual differences on N250 changes (in the Joe condition in Figure 3) related to their behavioral learning effects in response accuracy (in Table 1)? What other factors may contribute to such individual differences?
3. Is there any individual difference concerning the speed of N250 changes? For example, an individual may increase N250 earlier than another individual during the experiment.

Minor points:

1. It is not clear how the Other face condition was defined in this study. In Tanaka et al. (2006), one face was selected from 10 others faces to define the Other face condition (therefore, the amount of trials for ERP analysis was equal for the three conditions). Was the Other face condition defined in the same way in the present study?
2. In line 26 of page 5, to my knowledge, the N250 component is unusually distributed at occipito-temporal sites or inferior temporal sites but not occipito-central sites.

Decision letter (RSOS-202356.R2)

Dear Dr Sommer

On behalf of the Editor, I am pleased to inform you that your Stage 2 Replication submission RSOS-202356.R2 entitled "The N250 Event-related Potential as an Index of Face Familiarity: A Replication Study" has been accepted for publication in Royal Society Open Science subject to

minor revision in accordance with the referee suggestions. Please find the referees' comments at the end of this email.

The reviewers and Subject Editor have recommended publication, but also suggest some minor revisions to your manuscript. Therefore, I invite you to respond to the comments and revise your manuscript.

Please also ensure that all the below editorial sections are included where appropriate (a non-exhaustive example is included in an attachment):

- Ethics statement

- Data accessibility

If you wish to submit your supporting data or code to Dryad (<http://datadryad.org/>), or modify your current submission to dryad, please use the following link:
<http://datadryad.org/submit?journalID=RSOS&manu=RSOS-202356.R2>

- Competing interests

- Authors' contributions

- Acknowledgements

- Funding statement

Because the schedule for publication is very tight, it is a condition of publication that you submit the revised version of your manuscript within 7 days (i.e. by the 28-Apr-2021). If you do not think you will be able to meet this date please let me know immediately.

- 1) A text file of the manuscript (tex, txt, rtf, docx or doc), references, tables (including captions) and figure captions. Do not upload a PDF as your "Main Document".
- 2) A separate electronic file of each figure (EPS or print-quality PDF preferred (either format should be produced directly from original creation package), or original software format)
- 3) Included a 100 word media summary of your paper when requested at submission. Please ensure you have entered correct contact details (email, institution and telephone) in your user account
- 4) Included the raw data to support the claims made in your paper. You can either include your data as electronic supplementary material or upload to a repository and include the relevant DOI within your manuscript
- 5) Included your supplementary files in a format you are happy with (no line numbers, Vancouver referencing, track changes removed etc) as these files will NOT be edited in production

on behalf of Professor Chris Chambers (Registered Reports Editor, Royal Society Open Science)
openscience@royalsociety.org

Associate Editor Comments to Author (Professor Chris Chambers):

Two of the original reviewers from Stage 1 returned to assess the Stage 2 manuscript. Both are enthusiastic about the submission, judging that both Stage 1 primary criteria are met. The reviewers do offer some useful suggestions for final revision, including clarification of the methods, provision of open materials (which in any case is an RSOS policy requirement), and suggestions for a couple of additional exploratory analyses (see Reviewer 3, main point 2 & 3). Please note that additional analyses that go beyond the approved protocol are not required, but the authors are welcome to conduct and report them as they see fit. Please also note that no particular justification is required for the replication target (see Reviewer 2, point 3) -- the publication of the original study is sufficient justification under the terms of the RSOS Replication policy -- however, again, the authors are invited to consider this point in a final revision.

Provided the authors are able to respond thoroughly to the points raised in a Minor Revision, final acceptance should be forthcoming without requiring further in-depth review.

Reviewers' comments to Author:

Reviewer: 2

Comments to the Author(s)

This is an interesting and generally well-written replication attempt, which did reproduce the main findings of Tanaka et al. (2006), with the important caveat that the time course of face learning was different.

I have some minor comments for consideration by the authors.

- 1) The pattern observed for the N250 for Own faces appeared slightly different from the Tanaka study. This is noted by the authors, but no explanation is provided. Judging from their Figures, it looks as if this N250 is triggered earlier than the N250 to Joe, and this could be mentioned. Some evidence for such a latency difference was also present in Tanaka et al., see their Figure 2, in particular over the right hemisphere, although I'm not sure that this was actually discussed in their article.
- 2) Could samples of the faces images used be included in this report? Tanaka et al. provided such samples in their Figure 1, and it would be informative if these images could in principle be compared.
- 3) Could some explanation be provided as to whether the Tanaka et al. study was selected among the many other N250 studies of face recognition as the target of this replication study? Because it was the first/most widely cited?
- 4) Typo in the abstract (line 13): spilt -> split

Reviewer: 3

Comments to the Author(s)

This study tried to replicate the work of Tanaka et al. (2006) which found that the N250 component increased for an experimentally familiarized face in the second half of the experiment relative to the first half. Although this study could not directly replicate the previous findings by dividing the trials into two equal halves, it did replicate similar N250 increase when comparing 1/3 vs. 2/3 trials suggesting that participants may learn faster in this study. The methodology was sound, the results were clear, and the explanations were convinced. However, since another aim of this study was to investigate the individual differences during the learning process, I suggest the authors to give more exploration and discussion of this part of results.

Main points:

1. What conclusion could be drawn based on the present individual data of the N250 results?

2. Are the individual differences on N250 changes (in the Joe condition in Figure 3) related to their behavioral learning effects in response accuracy (in Table 1)? What other factors may contribute to such individual differences?
3. Is there any individual difference concerning the speed of N250 changes? For example, an individual may increase N250 earlier than another individual during the experiment.

Minor points:

1. It is not clear how the Other face condition was defined in this study. In Tanaka et al. (2006), one face was selected from 10 other faces to define the Other face condition (therefore, the amount of trials for ERP analysis was equal for the three conditions). Was the Other face condition defined in the same way in the present study?
2. In line 26 of page 5, to my knowledge, the N250 component is unusually distributed at occipito-temporal sites or inferior temporal sites but not occipito-central sites.

Author's Response to Decision Letter for (RSOS-202356.R2)

See Appendix C.

Decision letter (RSOS-202356.R3)

Dear Dr Sommer:

It is a pleasure to accept your manuscript entitled "The N250 Event-related Potential as an Index of Face Familiarity: A Replication Study" in its current form for publication in Royal Society Open Science.

on behalf of Professor Chris Chambers (Subject Editor)
openscience@royalsociety.org

Appendix A

Dear Dr. Chambers,

Thank you and the reviewers for your response to our submission.

Below we address the points raised by the reviewers.

Sincerely

Werner Sommer

Reviewer: 1

In the proposed experimental plan, I did not see a data analysis section. It is not stated how the ERP components (e.g, P1, N170, N250) be identified and analyzed. Please indicate whether there are plans to make the data available.

Response: Two paragraphs about data analysis have been added at the end of section 2.4. (marked in red).

As stated on Page 6, The raw EEG recordings in Brain Vision format, scripts for preprocessing of the raw recordings, and the code for generating results will be made available.

Reviewer: 2

"At first, participants were familiarized with the Joe or Jane target face. Although this part is crucial for the process of learning, the original description by Tanaka et al. (2006) [6] is a bit vague: "Subjects were then shown the target Joe (Jane) face, and asked to study it." (P. 1490). Thus, we decided to present the Joe (Jane) face on the screen right before the experiment, for 10 to 60 seconds depending on the participant's needs. Although it limits the possibility to analyze the initial phase of face learning, we decided to stick with the (presumably) original procedure."

I agree with the authors that details of the initial learning phase are crucial for any replication of a learning-based effect, and wonder why they did not contact the lead author of the original study (Jim Tanaka) for clarification. I am sure that Jim would be happy to provide them with all details required to match their learning phase with that of the original study. If data collection has not yet commenced, I recommend that the authors contact Dr Tanaka. If data collection has already happened, they should still do so and refer to any differences in learning procedures in their article.

Response: We have tried several times to contact Dr. Tanaka, but so far we have received no reply. However, following the advice of Reviewer 2, we will try again.

Appendix B

Dear Prof. Chambers,

We are delighted about the acceptance of our manuscript pending minor revisions! We thank you, the reviewers and the editorial team for the pleasant experience of submitting to RSOS!

Below please find the description how we dealt with the remaining requests. The formal requirements should all have been met. New or altered text in the manuscript is marked yellow.

Cordially, on behalf also of my coauthors
Werner Sommer

Associate Editor Comments to Author (Professor Chris Chambers):

Two of the original reviewers from Stage 1 returned to assess the Stage 2 manuscript. Both are enthusiastic about the submission, judging that both Stage 1 primary criteria are met. The reviewers do offer some useful suggestions for final revision, including clarification of the methods, provision of open materials (which in any case is an RSOS policy requirement), and suggestions for a couple of additional exploratory analyses (see Reviewer 3, main point 2 & 3). Please note that additional analyses that go beyond the approved protocol are not required, but the authors are welcome to conduct and report them as they see fit. Please also note that no particular justification is required for the replication target (see Reviewer 2, point 3) -- the publication of the original study is sufficient justification under the terms of the RSOS Replication policy -- however, again, the authors are invited to consider this point in a final revision.

Provided the authors are able to respond thoroughly to the points raised in a Minor Revision, final acceptance should be forthcoming without requiring further in-depth review.

Response: Thank you for this positive evaluation and the useful advice. Below we describe how we addressed the points raised by the reviewers.

Reviewers' comments to Author:

Reviewer: 2
Comments to the Author(s)

This is an interesting and generally well-written replication attempt, which did reproduce the main findings of Tanaka et al. (2006), with the important caveat that the time course of face learning was different.

I have some minor comments for consideration by the authors.

1) The pattern observed for the N250 for Own faces appeared slightly different from the Tanaka study. This is noted by the authors, but no explanation is provided. Judging from their Figures, it looks as if this N250 is triggered earlier than the N250 to Joe, and this could be mentioned. Some evidence for such a latency difference was also present in Tanaka et al., see their Figure 2, in particular over the right hemisphere, although I'm not sure that this was actually discussed in their article.

Response: We agree with the reviewer that there seems to be some latency shift in the N250 to Own faces and Joe faces. However, this shift might be due to the overlapping following positivity, which is much larger in case of the Own faces. This positivity might shift the N250 towards more positive values and – at the same time – reduce its peak latency as shown in the simulation included: Assuming the ERP around N250 actually consists of two peaks, the increased height of the right peak shift N250 latency to the left (earlier), and you need only a slight shift of the peak itself to change the negative dip into an inflection of the ERP slope (which we do see in all the plots at 230 ms). Because this explanation is speculative and cannot be linked to any psychological construct, we prefer not to go into it in the manuscript but would be willing to do so if advised by the editor.

```
In[194]:=  $\sigma = 0.05; \mu_1 = 0.25; \mu_2 = 0.4; \mu_3 = 0.38;$ 
```

$$f1[x_] := e^{-\frac{(x-\mu_1)^2}{2\sigma^2}} + e^{-\frac{(x-\mu_2)^2}{2\sigma^2}};$$

$$f2[x_] := e^{-\frac{(x-\mu_1)^2}{2\sigma^2}} + 2e^{-\frac{(x-\mu_2)^2}{2\sigma^2}};$$

$$f3[x_] := e^{-\frac{(x-\mu_1)^2}{2\sigma^2}} + 2e^{-\frac{(x-\mu_3)^2}{2\sigma^2}};$$

```
In[198]:= Plot[{f1[x], f2[x], f3[x]}, {x, 0.1, .6}, PlotRange -> All, PlotLegends -> "Expressions"]
```

2) Could samples of the faces images used be included in this report? Tanaka et al. provided such samples in their Figure 1, and it would be informative if these images could in principle be compared.

Response: We now include a figure with some sample faces (Fig. 1).

3) Could some explanation be provided as to whether the Tanaka et al. study was selected among the many other N250 studies of face recognition as the target of this replication study? Because it was the first/most widely cited?

Response: As mentioned in the article the original study is an important study in the field and therefore warrants replication.

4) Typo in the abstract (line 13): spilt -> split

Response

It has been corrected

Reviewer: 3

Comments to the Author(s)

This study tried to replicate the work of Tanaka et al. (2006) which found that the N250 component increased for an experimentally familiarized face in the second half of the experiment relative to the first half. Although this study could not directly replicate the previous findings by dividing the trials into two equal halves, it did replicate similar N250 increase when comparing 1/3 vs. 2/3 trials suggesting that participants may learn faster in this study. The methodology was sound, the results were clear, and the explanations were convinced. However, since another aim of this study was to investigate the individual differences during the learning process, I suggest the authors to give more exploration and discussion of this part of results.

Response: Thank you for the positive evaluation of our study. A deeper analysis of the individual differences across the learning process is a perspective that we will take in future research. We have added a sentence about this perspective in the discussion. "Another interesting line of

research in the future might be a more sophisticated analysis of the participant's individual time courses of the development of N250 amplitude, as shown in Figure 4, especially in their relationship to face recognition abilities.”

Main points:

1. What conclusion could be drawn based on the present individual data of the N250 results?

Response: Please see response above

2. Are the individual differences on N250 changes (in the Joe condition in Figure 3) related to their behavioral learning effects in response accuracy (in Table 1)? What other factors may contribute to such individual differences?

Response: Please see response above. Indeed, it is our future aim to study such relationships but refrained to do so in the present report because it would go far beyond the replication.

3. Is there any individual difference concerning the speed of N250 changes? For example, an individual may increase N250 earlier than another individual during the experiment.

Response: Very good point that will hopefully be addressed in a future report.

Minor points:

1. It is not clear how the Other face condition was defined in this study. In Tanaka et al. (2006), one face was selected from 10 others faces to define the Other face condition (therefore, the amount of trials for ERP analysis was equal for the three conditions). Was the Other face condition defined in the same way in the present study?

Response: In our study we used all the other faces unlike in Tanaka. So the Other sample was indeed larger. We have made this clear by specifying the conditions in the sentence about averaging: " ERP waveforms were obtained by averaging over all electrodes within each ROI, per condition (Joe/Jane, Own, Other) and each half of the experiment (Fig. 2)."

2. In line 26 of page 5, to my knowledge, the N250 component is unusually distributed at occipito-temporal sites or inferior temporal sites but not occipito-central sites.

Response:

Thank you for your remark, it has been corrected to “occipito-temporal”

Appendix C

Dear Prof. Chambers,

We are delighted about the acceptance of our manuscript pending minor revisions! We thank you, the reviewers and the editorial team for the pleasant experience of submitting to RSOS!

Below please find the description how we dealt with the remaining requests. The formal requirements should all have been met. New or altered text in the manuscript is marked yellow.

Cordially, on behalf also of my coauthors
Werner Sommer

Associate Editor Comments to Author (Professor Chris Chambers):

Two of the original reviewers from Stage 1 returned to assess the Stage 2 manuscript. Both are enthusiastic about the submission, judging that both Stage 1 primary criteria are met. The reviewers do offer some useful suggestions for final revision, including clarification of the methods, provision of open materials (which in any case is an RSOS policy requirement), and suggestions for a couple of additional exploratory analyses (see Reviewer 3, main point 2 & 3). Please note that additional analyses that go beyond the approved protocol are not required, but the authors are welcome to conduct and report them as they see fit. Please also note that no particular justification is required for the replication target (see Reviewer 2, point 3) -- the publication of the original study is sufficient justification under the terms of the RSOS Replication policy -- however, again, the authors are invited to consider this point in a final revision.

Provided the authors are able to respond thoroughly to the points raised in a Minor Revision, final acceptance should be forthcoming without requiring further in-depth review.

Response: Thank you for this positive evaluation and the useful advice. Below we describe how we addressed the points raised by the reviewers.

Reviewers' comments to Author:

Reviewer: 2
Comments to the Author(s)

This is an interesting and generally well-written replication attempt, which did reproduce the main findings of Tanaka et al. (2006), with the important caveat that the time course of face learning was different.

I have some minor comments for consideration by the authors.

1) The pattern observed for the N250 for Own faces appeared slightly different from the Tanaka study. This is noted by the authors, but no explanation is provided. Judging from their Figures, it looks as if this N250 is triggered earlier than the N250 to Joe, and this could be mentioned. Some evidence for such a latency difference was also present in Tanaka et al., see their Figure 2, in particular over the right hemisphere, although I'm not sure that this was actually discussed in their article.

Response: We agree with the reviewer that there seems to be some latency shift in the N250 to Own faces and Joe faces. However, this shift might be due to the overlapping following positivity, which is much larger in case of the Own faces. This positivity might shift the N250 towards more positive values and – at the same time – reduce its peak latency as shown in the simulation included: Assuming the ERP around N250 actually consists of two peaks, the increased height of the right peak shift N250 latency to the left (earlier), and you need only a slight shift of the peak itself to change the negative dip into an inflection of the ERP slope (which we do see in all the plots at 230 ms). Because this explanation is speculative and cannot be linked to any psychological construct, we prefer not to go into it in the manuscript but would be willing to do so if advised by the editor.

In[194]:= $\sigma = 0.05; \mu_1 = 0.25; \mu_2 = 0.4; \mu_3 = 0.38;$

$$f_1[x_] := e^{-\frac{(x-\mu_1)^2}{2\sigma^2}} + e^{-\frac{(x-\mu_2)^2}{2\sigma^2}};$$

$$f_2[x_] := e^{-\frac{(x-\mu_1)^2}{2\sigma^2}} + 2e^{-\frac{(x-\mu_2)^2}{2\sigma^2}};$$

$$f_3[x_] := e^{-\frac{(x-\mu_1)^2}{2\sigma^2}} + 2e^{-\frac{(x-\mu_3)^2}{2\sigma^2}};$$

In[198]:= `Plot[{f1[x], f2[x], f3[x]}, {x, 0.1, .6}, PlotRange -> All, PlotLegends -> "Expressions"]`

2) Could samples of the faces images used be included in this report? Tanaka et al. provided such samples in their Figure 1, and it would be informative if these images could in principle be compared.

Response: We now include a figure with some sample faces (Fig. 1).

3) Could some explanation be provided as to whether the Tanaka et al. study was selected among the many other N250 studies of face recognition as the target of this replication study? Because it was the first/most widely cited?

Response: As mentioned in the article the original study is an important study in the field and therefore warrants replication.

4) Typo in the abstract (line 13): spilt -> split

Response

It has been corrected

Reviewer: 3

Comments to the Author(s)

This study tried to replicate the work of Tanaka et al. (2006) which found that the N250 component increased for an experimentally familiarized face in the second half of the experiment relative to the first half. Although this study could not directly replicate the previous findings by dividing the trials into two equal halves, it did replicate similar N250 increase when comparing 1/3 vs. 2/3 trials suggesting that participants may learn faster in this study. The methodology was sound, the results were clear, and the explanations were convinced. However, since another aim of this study was to investigate the individual differences during the learning process, I suggest the authors to give more exploration and discussion of this part of results.

Response: Thank you for the positive evaluation of our study. A deeper analysis of the individual differences across the learning process is a perspective that we will take in future research. We have added a sentence about this perspective in the discussion. "Another interesting line of

research in the future might be a more sophisticated analysis of the participant's individual time courses of the development of N250 amplitude, as shown in Figure 4, especially in their relationship to face recognition abilities.”

Main points:

1. What conclusion could be drawn based on the present individual data of the N250 results?

Response: Please see response above

2. Are the individual differences on N250 changes (in the Joe condition in Figure 3) related to their behavioral learning effects in response accuracy (in Table 1)? What other factors may contribute to such individual differences?

Response: Please see response above. Indeed, it is our future aim to study such relationships but refrained to do so in the present report because it would go far beyond the replication.

3. Is there any individual difference concerning the speed of N250 changes? For example, an individual may increase N250 earlier than another individual during the experiment.

Response: Very good point that will hopefully be addressed in a future report.

Minor points:

1. It is not clear how the Other face condition was defined in this study. In Tanaka et al. (2006), one face was selected from 10 others faces to define the Other face condition (therefore, the amount of trials for ERP analysis was equal for the three conditions). Was the Other face condition defined in the same way in the present study?

Response: In our study we used all the other faces unlike in Tanaka. So the Other sample was indeed larger. We have made this clear by specifying the conditions in the sentence about averaging: " ERP waveforms were obtained by averaging over all electrodes within each ROI, per condition (Joe/Jane, Own, Other) and each half of the experiment (Fig. 2)."

2. In line 26 of page 5, to my knowledge, the N250 component is unusually distributed at occipito-temporal sites or inferior temporal sites but not occipito-central sites.

Response:

Thank you for your remark, it has been corrected to “occipito-temporal”